# REVISIT MODEL ADAPTATION FROM PARAMETERS TO FEATURES

## ABSTRACT

In this paper, we focus on an intriguing question: *Can existing fine-tuning adapters, such as LoRA, trained on one model be effectively transferred to its parameter-wise variants?* To investigate this problem, we first examine the technical underpinnings of widely adopted parameter-efficient fine-tuning methods. Our theoretical analysis reveals that, due to the strong coupling between adaptation components and base weights, these methods are vulnerable to weight transformations, leading to unsatisfactory cross-model performance and potential model-specific overfitting. To alleviate this issue, we accordingly propose two alternatives, which pose the adaptation on the input and output features, respectively, with an explicit decoupling scheme. In this way, the adaptation components for an unseen base model can be modulated by its native parameters and thus exhibit more robust transferability. Notably, the proposed methods can serve as plug-and-play components with merely one-line code modifications required. Though extremely simple, extensive experiments across a variety of models and applications demonstrate that our method achieves comparable performance to existing counterparts on the source model and consistently outperforms them in cross-model transfer settings.

## 1 INTRODUCTION

Recent years have witnessed dramatic progressions in artificial general intelligence (AGI), fueled by foundation models in a variety of domains, such as vision Radford et al. (2021); Rombach et al. (2022), language Brown et al. (2020); Touvron et al. (2023a;b); Grattafiori et al. (2024); Bai et al. (2023a); Yang et al. (2024a); Liu et al. (2024a), audio Chu et al. (2024); Yang et al. (2023), and their intersections Bai et al. (2023b); Wang et al. (2024b); Liu et al. (2023a). In many practical cases, users seek to adapt these foundation models into specific domains or equip them with novel concepts Ruiz et al. (2023); Han et al. (2023); Kumari et al. (2023). To address these needs, a series of works are dedicated to effective model adaptation approaches, particularly parameter-efficient fine-tuning (PEFT) methods, which adapt pretrained models by modifying only a small subset of parameters or incorporating lightweight modules Lialin et al. (2023); Hu et al. (2022); Yeh et al. (2023); Yang et al. (2024b); Meng et al. (2024); Wang et al. (2024a); Liu et al. (2024b). For instance, low-rank adaptation (LoRA) Hu et al. (2022) freezes the pretrained weights and learns low-rank update matrices for each layer, yielding orders-of-magnitude fewer trainable parameters, *e.g.*, $\sim 10,000\times$ reduction, while matching or even exceeding full fine-tuning performance.

Although these PEFT methods have achieved remarkable progress in model adaptation and have become standard practice in both academic research and industrial deployment, when the underlying base model changes, it remains unclear whether existing adapters trained on one base model can be effectively transferred to a different model, and whether current methods achieve optimal performance in cross-model settings. These scenarios are practical since modern foundation models can be updated frequently to accommodate novel information and functionalities, *e.g.*, Stable Diffusion series models Rombach et al. (2022); Podell et al. (2023); Esser et al. (2024). However, since existing fine-tuners are typically designed on a per-model-per-adaptation basis, they have to be retrained on the original adaptation data for these cases, which is highly cumbersome if not infeasible at all.

To alleviate this dilemma, in this paper, we begin by exploring an intriguing question: *Can popular fine-tuners, such as LoRA, trained on a source model be effectively transferred to its parameter-wise variants*, *e.g.*, models within the same family? Our preliminary studies on diffusion models Ho et al. (2020); Nichol & Dhariwal (2021); Rombach et al. (2022) indicate that such transferred adapters can produce plausible images; however, as illustrated in Fig. 1, their performance remains suboptimal and leaves significant room for improvement.

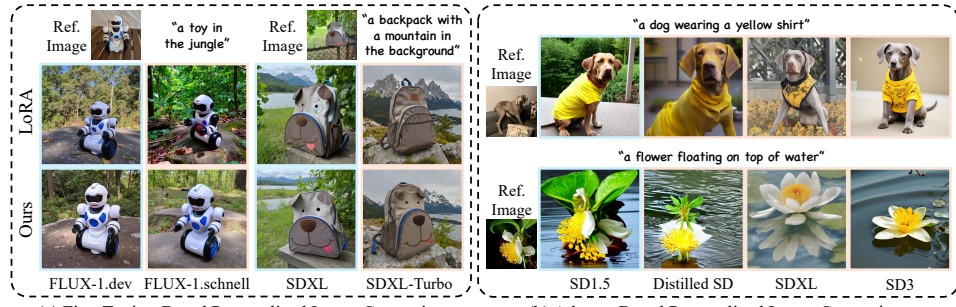

(a) Fine-Tuning-Based Personalized Image Generation  (b) Adapter-Based Personalized Image Generation

Figure 1: In this paper, we focus on the problem of cross-model adaptation, which concerns how well a fine-tuner or adapter trained on a source model generalizes to an unseen target model. We propose a simple yet effective strategy based on input/output feature adaptation, which consistently outperforms widely used parameter tuning approaches such as LoRA in this setting. Results of fine-tuning-based and adapter-based personalized image generation are shown here. Images generated by the source and target models are marked by the blue and orange borders, respectively.

We thus delve into the technical foundations of widely adopted fine-tuners and identify a key limitation: the adaptation components are tightly coupled with the specific weights of the base model, which fundamentally hinders their transferability across models. Specifically, consider the case of applying LoRA to a linear layer with a weight matrix $W \in \mathbb{R}^{c \times d}$, where $c$ and $d$ denote the input and output dimensions, respectively. Let $X \in \mathbb{R}^{n \times c}$ and $Y \in \mathbb{R}^{n \times d}$ represent the input and output matrices, where $n$ is the batch dimension. LoRA learns two low-rank matrices $A \in \mathbb{R}^{c \times r}$ and $B \in \mathbb{R}^{r \times d}$, where the rank $r \ll \min\{c, d\}$. For $Y = XW$, the forward propagation is updated as:

$$Y = XW + XAB = X(W + AB), \tag{1}$$

where the adaptation components $AB$ are applied onto the weight matrix. For a different target model with the weights $W'$, since the parameter spaces of the source and target can be different, the weight offsets trained on the source model may not be applicable to the target. In other words, directly transferring the weight offsets, i.e., $W' + AB$, may not be meaningful due to discrepancies between the parameter spaces of the source and target models.

The above analysis highlights the importance of decoupling the adaptation components from the base model's parameters. To this end, we aim at alternative spaces for these adaptation components to enable effective cross-model transfer of fine-tuners. In this paper, we turn to the input/output feature spaces of neural networks, motivated by the Platonic Representation Hypothesis Huh et al. (2024), which indicates that representations in AI models—particularly deep networks—tend to converge. Recent studies Yang et al. (2022); Xu et al. (2024); Pan et al. (2023) also suggest that, under certain circumstances, various network blocks can be treated as functionally equivalent, especially among those within the same architectural family. Furthermore, in many cases, the feature spaces across different models are exactly aligned; for instance, a range of text-to-image diffusion models Rombach et al. (2022); Luo et al. (2023a); Kim et al. (2024) utilize the same CLIP text encoder to process input prompts. Based on these insights, we hypothesize that the feature spaces of neural networks are more transferable across different models.

Correspondingly, we propose two fine-tuning strategies as shown in Fig. 3: rather than learning weight offsets, we perform adaptation directly in the input and output feature spaces, i.e., the spaces of $X$ and $Y$, respectively. One notable advantage of such designs lies in that the adaptation components are modulated by the native parameters of the base model, enabling more robust transferability across models. Moreover, these input/output adapters are not limited to LoRA. Instead, they can be seamlessly integrated into a wide spectrum of fine-tuning methods, requiring only one-line code modifications. Despite their simplicity, extensive experiments across diverse models and applications, including personalized and controllable image generation, architectural adaptation, and large language models, show that our methods achieve on-par performance with their counterparts on the source model, while consistently outperforming them in cross-model transfer scenarios. Our contributions can be summarized as follows:

- We figure out the underlying limitation of popular fine-tuning methods in terms of cross-model transferability and propose simple yet effective strategies to tackle the challenge;

- Leveraging the potential consistency of input and output feature spaces, we provide a theoretical analysis of the applicability and effectiveness of the proposed methods;
- We conduct extensive experiments across a range of models and applications, which demonstrate that the proposed methods can serve as plug-and-play components for a variety of fine-tuning strategies, consistently enhancing cross-model transferability while maintaining competitive performance on the source model.

## 2 RELATED WORKS

In this section, we review related works from two perspectives relevant to our proposed methods: (1) parameter-efficient fine-tuning and (2) cross-model fine-tuner transfer.

### 2.1 PARAMETER-EFFICIENT FINE-TUNING

Given a base model, parameter-efficient fine-tuning (PEFT) techniques Xu et al. (2023) aim to adapt it to a different domain by updating only a small subset of its parameters, which significantly reduces memory overhead during fine-tuning and is particularly advantageous in low-resource scenarios, such as for end users. Popular PEFT methods include adapter tuning Hu et al. (2022); Sung et al. (2022), prompt tuning Lester et al. (2021), and prefix tuning Li & Liang (2021). In this paper, we mainly focus on those based on adapter tuning, given its widespread usage in a series of scenarios, such as visual generation Ruiz et al. (2023); Tan et al. (2024); Luo et al. (2023b), language understanding Mao et al. (2025), and multi-modal tasks Zhou et al. (2024).

A representative method in this category, LoRA Hu et al. (2022), observes that model updates often lie in a low-rank subspace and introduces low-rank adaptation modules to achieve parameter efficiency. Recent studies have explored alternative weight decomposition strategies to enhance parameter-efficient fine-tuning, including but not limited to magnitude-direction decomposition in DoRA Liu et al. (2024b), singular value decomposition in PiSSA Meng et al. (2024), CorDA Yang et al. (2024b), and MiLoRA Wang et al. (2024a), frequency-based decomposition in FourierFT Gao et al. (2024), Hadamard-product-based LoHa Hyeon-Woo et al. (2021), and Kronecker-product-based LoKr Yeh et al. (2023), and others Qiu et al. (2023); Liu et al. (2023b); Kopiczko et al. (2023); Liu et al. (2022).

In this paper, we propose fine-tuning strategies that are orthogonal to existing methods and can serve as plug-and-play components, easily integrated with them to enhance performance in cross-model transfer scenarios. Please refer to Sec. 3 for the methodology and Sec. 5 for the experimental results.

### 2.2 CROSS-MODEL FINE-TUNER TRANSFER

In the literature, we find that there are works tackling the challenge of cross-model fine-tuner transfer from various perspectives. Lin *et al.* Lin et al. (2025) reveal that the weight offsets through fine-tuning are transferable to various base models. However, our theoretical and experimental studies suggest the limitations of such direct transfer. Wang *et al.* Wang et al. (2024d) and Ran *et al.* Ran et al. (2024) propose training-based solutions to adapt fine-tuners onto various base models, which are different from the training-free perspective in this paper. Most recently, Farhadzadeh *et al.* Farhadzadeh et al. (2025) introduce LoRA-X. This approach exploits subspace consistency between source and target weights to achieve training-free cross-model fine-tuner adaptation. Nevertheless, the assumption of consistent parameter subspaces remains strong and may limit generalizability. Moreover, LoRA-X constrains weight updates to a bottleneck matrix of shape $r \times r$ or even its diagonal elements, which inevitably limits its adaptation capacity on the source model compared to LoRA counterparts with the same forward computational budget.

## 3 METHODS

### 3.1 MOTIVATIONS

As mentioned in Sec. 1, the parameter spaces of various models can be different. To verify this claim, in Fig. 2, we visualize the average pairwise cosine similarity of the common parameters among

Stable Diffusion v1.5 Rombach et al. (2022) and its variants Luo et al. (2023b); Kim et al. (2024). Results show that, despite originating from the same source model, the fine-tuning process shifts the parameter spaces of these variants, reducing their robustness to weight offsets trained on the original model Lin et al. (2025).

Fortunately, we notice that, although parameters are different, all these models take features predicted by a common variational autoencoder and CLIP text model as input, suggesting the feasibility of posing the adaptation to the input space. Similarly, if the outputs exhibit consistency for various inputs, it would be beneficial to conduct adaptation in the output space.

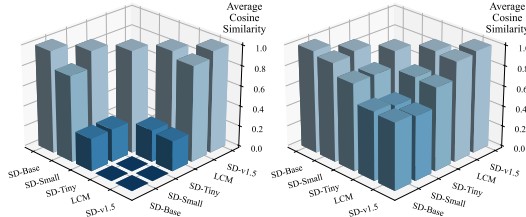

(a) 20% Layers with the Lowest Cosine Similarity     (b) All Layers

Figure 2: Pairwise similarity measurements among variants of Stable Diffusion (SD) v1.5.

### 3.2 INPUT/OUTPUT ADAPTATION

Motivated by the above analysis, we introduce the technical details of the proposed input/output adaptation strategies in this part. Still taking LoRA on a linear transformation layer as an example, we perform adaptation on its input/output with a couple of learnable low-rank matrices. The input and output adaptation can be formulated as:

$$Y = (X + XAB)W \quad \text{and} \quad Y = XW + XWAB, \tag{2}$$

respectively. Fig. 3(upper) provides an illustration of their workflows, and Fig. 3(below) presents the corresponding PyTorch-style Paszke (2019) pseudo codes. Notably, the proposed strategies require only one-line modifications to existing LoRA implementations, highlighting their ease in practical deployment. In terms of efficiency, when the input and output feature dimensions are equal, *i.e.*, $c = d$, these modifications incur no additional computational overhead compared to vanilla LoRA.

After training, when we want to deploy the adapter $AB$ on another model, we can simply replace $W$ in Eq. 2 with the new parameter $W'$. In contrast to vanilla LoRA, which uses static update components $XAB$ as shown in Eq. 1, our methods generate output offsets conditioned on the deployed base weights, thereby benefiting cross-model adaptability. Please refer to Sec. 4 for formal analyses.

### 3.3 PLUG-AND-PLAY COMPONENTS

It is worth noting that, although LoRA is used as an illustrative example to present the proposed workflows, the underlying insights are broadly applicable to a range of weight adaptation methods, including full fine-tuning. This part elaborates on how this is achieved. Closely examining Eq. 2, the computation can be rewritten as:

$$Y = X(I + AB)W \quad \text{and} \quad Y = XW(I + AB). \tag{3}$$

Eq. 3 indicates that, in the case of LoRA, the proposed input/output adaptation can be interpreted as injecting low-rank adapters into a linear transformation layer initialized with an identity matrix.

Therefore, the essence of the proposed strategies lies in keeping the base weights fixed while applying the adaptation to an identity transformation $I$, which is inserted at the beginning or end of the current layer in the base model, corresponding to input and output adaptation, respectively. This principle implies that any adaptation function $\Phi_\theta$, with learnable parameters $\theta$, can be applied to the identity matrix $I$ in the same manner as it would traditionally be applied to the base weight matrix $W$ and initialize the parameters $\theta$. More broadly, for a base layer $f$ parameterized by $W$, the proposed input and output adaptation strategies can be formalized as Eqs. 4b and 4c, respectively.

$$Y = f(X; \Phi_\theta(W)), \quad (4a) \quad Y = f(f(X; \Phi_\theta(I)); W), \quad (4b) \quad Y = f(f(X; W); \Phi_\theta(I)). \quad (4c)$$

We also supplement the expression of popular weight adapters in Eq. 4a for a clear comparison. If $f$ is a linear operator, *e.g.*, a linear transformation or convolution, the adaptation components can be absorbed into the base weights—an inherent property of most PEFT methods—ensuring that the fine-tuned model maintains the same computational cost as the base model at deployment.

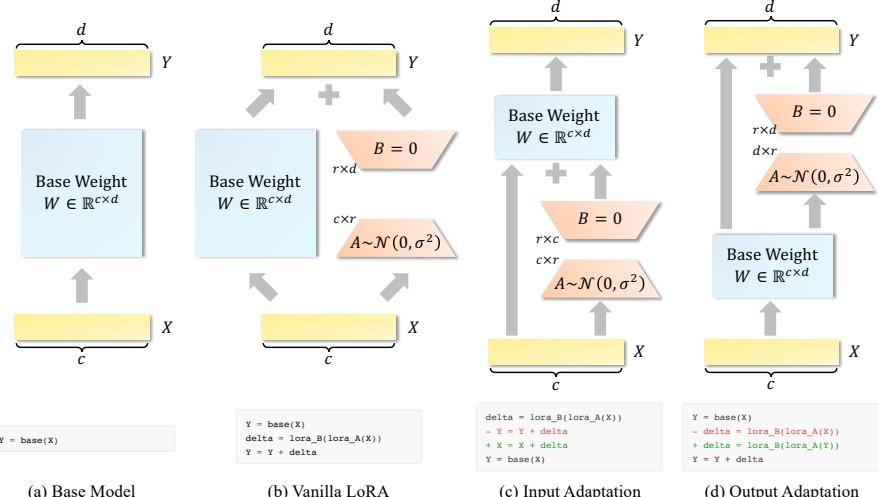

(a) Base Model     (b) Vanilla LoRA     (c) Input Adaptation     (d) Output Adaptation

Figure 3: Illustrative and algorithmic workflows of the base model, vanilla LoRA, and the proposed input/output adaptation. Blue modules are frozen in training, while orange modules are learnable and require gradient during adaptation.

## 4 THEORETICAL ANALYSIS

This section presents theoretical evidence for the effectiveness of the proposed input/output adaptation schemes in cross-model fine-tuner transfer. Specifically, we examine two scenarios, one assuming consistency in relative representation similarity and the other assuming consistency in model functionalities between the source and target. The former assumption is widely employed to design auxiliary supervision signals that accelerate training by aligning feature representations Yu et al. (2024), whereas the latter entails a stronger requirement, typically satisfied when the input and output spaces of the source and target models are closely aligned, *e.g.*, in diffusion models that leverage CLIP for conditional encoding.

For the former, consider one linear layer, and the source and target models can be written as $Y = XW$ and $Y' = XW'$, respectively. $x_1$ and $x_2$ are two feature vectors in the input space of the source model, and $y_1$ and $y_2$ are the corresponding output vectors with the parameter $W$. $x_1'$, $x_2'$, $y_1'$, and $y_2'$ are defined for the target model, similarly. Assume that the cosine similarity between $x_1$ and $x_2$, denoted as $\text{sim}(x_1, x_2)$, is equal to that between $x_1'$ and $x_2'$, *i.e.*, $\text{sim}(x_1, x_2) = \text{sim}(x_1', x_2')$, and that $s_0 := \text{sim}(y_1, y_2) = \text{sim}(y_1', y_2')$. The three adaptation strategies in Eqs. 1 and 3 are considered. The adapters $AB$ are trained on the source model and transferred to the target. Assume that the adaptation components are $L2$-regularized, *i.e.*, $\|AB\|_2^2 \leq \varepsilon$. We are interested in the similarity distance between output vectors produced under various adaptation strategies, *i.e.*, $|\text{sim}(\hat{y}_1, \hat{y}_2) - \text{sim}(\hat{y}_1', \hat{y}_2')|$, where $\hat{\cdot}$ denotes outputs from models after adaptation. The theoretical results concerning the three strategies are summarized as follows:

**Proposition 1.** *After applying the vanilla LoRA shown in Eq. 1 on both source and target models, the similarity distance between output vectors satisfies* $|\text{sim}(\hat{y}_1, \hat{y}_2) - \text{sim}(\hat{y}_1', \hat{y}_2')| \leq (1 + s_0)\Big(\frac{\|\mathbf{x}_1\| \varepsilon}{\|\mathbf{x}_1 W\|} + \frac{\|\mathbf{x}_2\| \varepsilon}{\|\mathbf{x}_2 W\|} + \frac{\|\mathbf{x}_1'\| \varepsilon}{\|\mathbf{x}_1' W'\|} + \frac{\|\mathbf{x}_2'\| \varepsilon}{\|\mathbf{x}_2' W'\|}\Big) + O(\varepsilon^2)$.

**Proposition 2.** *After applying the input or output adaptation shown in Eq. 3 on both source and target models, the similarity distance between output vectors satisfies* $|\text{sim}(\hat{y}_1, \hat{y}_2) - \text{sim}(\hat{y}_1', \hat{y}_2')| \leq 4(1 + s_0)\varepsilon + O(\varepsilon^2)$.

The proof and further analyses, including those for the latter scenario, are provided in the appendix. Intuitively, the above propositions imply that the upper bound of the similarity distance induced by the proposed input/output adapters is independent of the base parameters of the source and target models, suggesting a more robust preservation of relative relationships during adaptation transfer.

Revisiting Eqs. 1 and 3, we observe that the proposed input/output adapters apply linear transformations to the base weights, enabling a broader range of operations—including rotation and scaling—beyond the pure offset/translation applied in vanilla LoRA. Consequently, in many practical scenarios, the proposed methods also lead to improved fine-tuning performance on the source models.

| Model & Metric | | SD-v1.5 | | | SD-v3.5 | | | FLUX | | | SD-XL | | | SD-XL (Cross-Att.) | | |
|---|---|---|---|---|---|---|---|---|---|---|---|---|---|---|---|---|
| | | C-T | C-I | D-I | C-T | C-I | D-I | C-T | C-I | D-I | C-T | C-I | D-I | C-T | C-I | D-I |
| Source Model | LoRA | **.294** | .770 | .578 | .307 | .787 | .652 | .298 | .801 | .679 | .293 | **.815** | **.694** | **.289** | .810 | **.673** |
| | Input Ada. | .293 | .777 | .587 | **.308** | .785 | .645 | **.299** | .797 | .666 | .299 | .806 | .671 | .285 | **.815** | .663 |
| | Output Ada. | **.294** | **.790** | **.626** | .306 | **.794** | **.665** | .296 | **.808** | **.685** | **.300** | .811 | .690 | **.289** | .806 | .664 |
| Target Model | LoRA | .289 | .785 | .608 | .301 | .780 | .642 | **.308** | .773 | .626 | .309 | .752 | .592 | **.307** | .743 | .566 |
| | Input Ada. | .288 | .792 | .618 | **.302** | .782 | .642 | **.308** | .784 | .627 | **.316** | .759 | .592 | .296 | **.783** | **.622** |
| | Output Ada. | **.294** | **.793** | **.634** | .298 | **.794** | **.659** | .306 | **.789** | **.639** | .309 | **.762** | **.614** | **.307** | .750 | .576 |

Table 1: Performance in same-model and cross-model settings for vanilla LoRA and the proposed input/output adaptation strategies on the DreamBooth benchmark.

## 5 EXPERIMENTS

In this section, we experimentally demonstrate the effectiveness of the proposed input/output adaptation techniques. Sec. 5.1 presents our main results of applying the input/output adaptation on top of LoRA and a series of PEFT methods in a variety of tasks, including personalized and controllable image generation, architectural adaptation, and large language models. Section 5.2 presents a further exploration of the properties of the proposed methods.

### 5.1 MAIN RESULTS

**Fine-Tuning-Based Personalized Image Generation.** In personalized image generation—also known as subject-driven image generation—the goal is to produce images that follow text prompts while faithfully preserving the identity and appearance of subjects from user-provided images Gal et al. (2022). A line of studies addresses this challenge by fine-tuning pre-trained text-to-image diffusion models to capture the appearance of given subjects, such as DreamBooth Ruiz et al. (2023) and many following works Kumari et al. (2023); Chen et al. (2023a). PEFT techniques are widely adopted in this field to facilitate this fine-tuning process.

In this paper, we conduct experiments on the popular codebase provided by the Diffusers library von Platen et al. (2022) and build a variety of PEFT adapters, including LoRA Hu et al. (2022), DoRA Liu et al. (2024b), LoHa Hyeon-Woo et al. (2021), and LoKr Yeh et al. (2023), upon multiple popular text-to-image diffusion models, including Stable Diffusion 1.5 Rombach et al. (2022) (SD-v1.5), Stable Diffusion XL Podell et al. (2023) (SD-XL), Stable Diffusion 3.5-Large Esser et al. (2024) (SD-v3.5), and FLUX 1.dev Labs (2024). For cross-model fine-tuner transfer, we consider their corresponding time-distilled variants, *i.e.*, Latent Consistency Model Luo et al. (2023a), SDXL-Turbo, SD3.5-Turbo Sauer et al. (2024), and FLUX 1.schnell Labs (2024), respectively, as the target models.

Following common practice, we evaluate performance on the DreamBooth dataset Ruiz et al. (2023), which includes 30 subjects, each represented by 4 to 6 images. For each subject, the base diffusion model is fine-tuned using the provided images and the prompt "a photo of a [c]", where [c] denotes the subject's class name. The fine-tuned model is then evaluated using 25 diverse textual prompts, each repeated across 5 random seeds. There are $30 \times 25 \times 5 = 3750$ test cases in total. For each case, we evaluate three metrics: C-I, D-I, and C-T. C-I and D-I assess the average cosine similarity between features of the generated images and the source subject images, extracted by the CLIP image encoder Radford et al. (2021) and the ViT-S/16 DINO encoder Caron et al. (2021), respectively. C-T measures the cosine similarity between the CLIP image features of the generated images and the CLIP text features of the editing prompts. The average results across all the test cases are summarized in Tabs. 1 and 2.

Through results, we observe that the proposed input/output adaptation methods achieve performance on par with baseline methods on the source model, while consistently outperforming them in cross-model transfer setups in terms of CLIP-I and DINO-I. While our methods may yield slightly lower CLIP-T scores in some cases, this reflects the inherent trade-off between appearance preservation and prompt fidelity commonly observed in subject-driven generation Huang et al. (2024); Hoang et al. (2025). Specifically, as training progresses, the diffusion model tends to overfit to the subject images, diminishing its responsiveness to text prompts. From this perspective, higher image consistency potentially indicates better adaptability and generalization of the fine-tuning methods. Qualitative comparisons in Fig. 1 align with this analysis. Please refer to the appendix for more visual results.

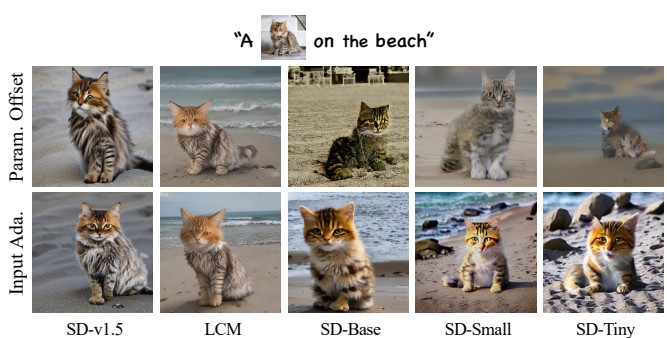

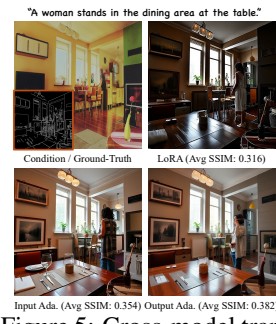

Figure 4: Qualitative results of adapter-based personalized image generation. Compared to transferring weight offsets Lin et al. (2025), the proposed input adapter achieves better cross-model transferability. SD-v1.5 is the source model, and others are unseen target models.

Figure 5: Cross-model transfer results from FLUX-1.dev to FLUX-1.schnell of OminiControl under the Canny edge condition.

| PEFT Method & Metric | | LoRA | | | DoRA | | | LoHa | | | LoKr | | |
|---|---|---|---|---|---|---|---|---|---|---|---|---|---|
| | | C-T | C-I | D-I | C-T | C-I | D-I | C-T | C-I | D-I | C-T | C-I | D-I |
| Source Model | Param. Ada. | .298 | .801 | .679 | **.303** | .801 | .672 | .275 | .819 | .697 | .289 | .818 | .704 |
| | Input Ada. | **.299** | .797 | .666 | **.303** | .803 | .676 | .279 | **.820** | **.700** | **.291** | .810 | .682 |
| | Output Ada. | .296 | **.808** | **.685** | .300 | **.810** | **.689** | .281 | .816 | .685 | .288 | **.820** | **.705** |
| Target Model | Param Ada. | **.308** | .773 | .626 | **.314** | .772 | .612 | .288 | .797 | .658 | **.305** | .785 | .625 |
| | Input Ada. | **.308** | .784 | .627 | .311 | .775 | .613 | **.294** | **.810** | **.683** | .304 | .786 | .630 |
| | Output Ada. | .306 | **.789** | **.639** | .308 | **.781** | **.633** | **.294** | .804 | .664 | .302 | **.797** | **.659** |

Table 2: Performance in same-model and cross-model settings for various PEFT methods and the proposed input/output adaptation strategies built upon each of them on the DreamBooth benchmark. The source and target models are FLUX-1.dev and FLUX-1.schnell, respectively.

**Adapter-Based Personalized Image Generation.** The fine-tuning-based personalized image generation methods mentioned above require a training process for each individual subject, which results in limited flexibility and efficiency in practice. To address this drawback, a series of approaches are dedicated to fine-tuning-free schemes Li et al. (2024); Shi et al. (2023); Chen et al. (2023b); Wang et al. (2024c); Wei et al. (2023); Ye et al. (2023). The key idea is to tame an adapter that maps a subject image into the conditional space of a diffusion model, enabling it to handle arbitrary subject images during inference in a feed-forward manner.

Although effective, it remains unclear whether an adapter trained on one diffusion model can be directly applied to a different target model. Empirically, we find that existing methods generally fail in this setting. In particular, when the source and target models have different dimensionalities, the adapter often cannot be loaded at all due to misaligned feature spaces. By contrast, our methods alleviate this issue by leveraging the consistent input space across various text-to-image models. Specifically, although different diffusion models may have distinct intermediate conditional spaces, many of them share a common input space. For instance, CLIP text features are used as input across a range of modern text-to-image models, including SD-v1.5 Rombach et al. (2022) and its distilled variants Kim et al. (2024), SD-XL Podell et al. (2023), SD-3.5 Esser et al. (2024), *etc*.

Based on this analysis, we apply the proposed input adaptation method to train the adapter on SD-v1.5 following ELITE. Wei et al. (2023) and evaluate its cross-model transferability on the remaining models. The benchmarks and evaluation protocols follow those described above. Results shown in Tabs. 3 and 4 demonstrate the effectiveness and superiority of our method in this setup. Figs. 1 and 4 present some qualitative visualizations. Additional visual comparisons can be found in the appendix.

**Controllable Image Generation.** A variety of works focus on training additional adapters to inject conditions beyond textual prompts as input to a pre-trained text-to-image diffusion model Zhang et al. (2023); Mou et al. (2023). We experiment with OminiControl Tan et al. (2024), a popular FLUX-based framework for controllable image generation that incorporates low-rank adapters into additional conditional branches, and explore the transferability of these adapters from FLUX-1.dev to FLUX-1.schnell Labs (2024). In Fig. 5, we present results under the Canny edge condition and report average SSIM scores over 5,000 validation images from COCO2017 Lin et al. (2014). Our input/output adaptation methods enhance visual results and deliver notable performance improvements.

| Target Model & Metric | SD-v1.5 | | | LCM | | | SD-Base | | | SD-Small | | | SD-Tiny | | |
|---|---|---|---|---|---|---|---|---|---|---|---|---|---|---|---|
| | C-T | C-I | D-I | C-T | C-I | D-I | C-T | C-I | D-I | C-T | C-I | D-I | C-T | C-I | D-I |
| Param. Ada. | .295 | **.782** | **.679** | .289 | .747 | .539 | .297 | .663 | .359 | .286 | .645 | .323 | .283 | .630 | .323 |
| Input Ada. | **.302** | .775 | .668 | **.301** | **.768** | **.596** | **.312** | **.740** | **.540** | **.308** | **.715** | **.518** | **.289** | **.671** | **.466** |

Table 3: Quantitative results of adapter-based personalized image generation. Trained on SD-v1.5, the adapter by our method achieves superior transferability to various target models.

| Metric | C-T | C-I | D-I |
|---|---|---|---|
| SD-XL | .294 | .766 | .580 |
| SD-v3.5 | .323 | .748 | .573 |

Table 4: Cross-model results of our input adaptation on models with large architectural gaps to the source model SD-v1.5.

| Metric | Img. Acc. | Text Acc. | Two Obj. | Pos. | Cnt. | Sin. Obj. | Col. | Col. Attr. | Overall |
|---|---|---|---|---|---|---|---|---|---|
| Param. Ada. | 65.73 | 79.57 | 84.60 | 21.25 | 64.06 | **99.06** | **81.38** | 51.50 | 0.670 |
| Input Ada. | **67.13** | 81.01 | **87.37** | **24.75** | **74.69** | 98.44 | 78.72 | 47.50 | **0.686** |
| Output Ada. | 66.00 | **81.74** | 84.85 | 21.00 | 67.81 | 98.75 | 79.52 | **52.00** | 0.673 |

Table 5: GenEval results by transferring the components for architectural adaptation on FLUX-Dev2Pro to FLUX-1.dev.

**Architectural Adaptation.** Recent works Liu et al. (2024d;c) find that it is feasible to accelerate pretrained diffusion transformers by replacing the attention modules with their efficient alternatives like neighborhood attention Hassani et al. (2023). After distillation-based adaptation, the new architecture can achieve on-par performance with the original one while enjoying better inference efficiency.

Nevertheless, recent findings Shi (2024) show that for guidance-distilled models such as FLUX-1.dev Labs (2024), direct adaptation often yields suboptimal results due to the entanglement introduced by classifier-free guidance Ho & Salimans (2022). For FLUX, it is recommended to conduct adaptation on its "undistilled" version Flux-Dev2Pro Ashen0209 (2024) and transfer the adapters to FLUX-1.dev during inference.

We thus apply the proposed input/output adaptation methods to this setup. The GenEval metrics Ghosh et al. (2023) shown in Tab. 5 demonstrate the advantages of the input adapter in this case. Output adaptation achieves performance comparable to the baseline of transferring weight offsets Lin et al. (2025). We speculate that it is due to a mismatch in output distributions—with and without classifier-free guidance.

**Large Language Model.** We further evaluate the proposed methods on widely used benchmarks for large language model (LLM) fine-tuning. Specifically, we use CommonsenseQA Talmor et al. (2018), MetaMath Yu et al. (2023), and Code-Feedback Zheng et al. (2024) as training datasets to assess capabilities in commonsense reasoning, mathematical problem-solving, and coding, respectively. The evaluation is conducted on the eight sub-tasks of CommonsenseQA, GSM8K Cobbe et al. (2021), and both HumanEval Chen et al. (2021) and MBPP Austin et al. (2021) corresponding to the three domains. We use Llama-3.2-1B Grattafiori et al. (2024) as the source model and its instruction-tuned variant, Llama-3.2-1B-Instruct, as the target model. Detailed setup here is provided in the appendix.

As shown in Tab. 6, input adaptation outperforms output adaptation and vanilla LoRA in this setting. We hypothesize that it is due to the similarity in input distributions, whereas the instruction-tuned target model tends to produce output styles that differ from those of the source model.

## 5.2 EMPIRICAL STUDIES

**Conversion from Vanilla LoRA.** We are curious about an interesting question: *Can a pre-trained vanilla LoRA be converted into the proposed input/output adapters, enabling cross-model transfer without the need for training new adapters?* We find that it is theoretically feasible. Taking the output adaptation in Eq. 3(right) as an example, a vanilla LoRA in Eq. 1 satisfies:

$$Y = X(W + AB) = XW(I + W^{\dagger}AB), \qquad (5)$$

where $^{\dagger}$ denotes the pseudoinverse and $WW^{\dagger} = I$. Since $A$ and $B$ are rank-$r$ matrices, the overall rank of the term $W^{\dagger}AB$ is at most $r$, which can also be factorized into two low-rank matrices via singular value decomposition, *i.e.*, there exist $A'$ and $B'$ satisfying $A'B' = W^{\dagger}AB$. $A'$ and $B'$ can serve as the parameters in the proposed output adapters.

Unfortunately, our experiments show that it may result in subpar performance. As illustrated in Fig. 6, transferring LoRA trained on FLUX-1.dev to FLUX-1.schnell Labs (2024) in this manner leads to diminished subject consistency and noticeably degraded image quality. These results indicate that the

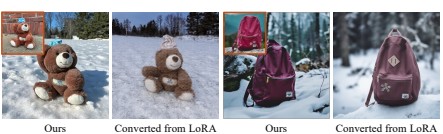

Figure 6: It is *infeasible* to convert a LoRA model pre-trained on FLUX-1.dev into the proposed input/output adaptation form via Eq. 5 and then apply it to FLUX-1.schnell.

Figure 7: The proposed input/output adaptation strategy can be integrated with existing PEFT methods, achieving the best of both worlds on certain metrics.

|  | Task | BoolQ | PIQA | SIQA | HellaSwag | WinoGrande | ARC-e | ARC-c | OBQA | Avg. | GSM8K | HumanEval | MBPP |
|---|---|---|---|---|---|---|---|---|---|---|---|---|---|
| Source Model | LoRA | **62.42** | 75.57 | 71.65 | **68.61** | 68.51 | 65.95 | 50.68 | 67.20 | 66.32 | 25.47 | 21.3 | 38.1 |
|  | Input Ada. | 62.14 | **76.17** | 71.65 | 63.75 | **69.22** | 68.98 | 51.28 | 66.20 | 66.17 | 23.96 | **23.8** | 37.0 |
|  | Output Ada. | 62.26 | 74.37 | **72.16** | 67.40 | 67.17 | **69.70** | **52.56** | **67.40** | **66.63** | **26.76** | 21.3 | **38.9** |
| Target Model | LoRA | 61.87 | 60.01 | 65.56 | 47.86 | 60.62 | 68.10 | 51.02 | 60.00 | 59.38 | 7.81 | 34.1 | 49.2 |
|  | Input Ada. | 61.96 | **69.26** | **69.70** | **54.20** | **64.96** | **69.53** | **52.65** | **65.80** | **63.51** | 16.07 | **42.1** | **49.7** |
|  | Output Ada. | **62.17** | 64.69 | 65.05 | 50.95 | 61.80 | 68.10 | 50.94 | 60.80 | 60.56 | **22.59** | 36.6 | 48.4 |

Table 6: Performance in same-model and cross-model settings for vanilla LoRA and the proposed input/output adaptation strategies in commonsense reasoning, mathematical problem-solving, and coding. The source and target models are Llama-3.2-1B and Llama-3.2-1B-Instruct, respectively.

strong coupling between adaptation components and base weights in vanilla LoRA is intrinsic and difficult to resolve through post-training strategies.

**Integration with Vanilla LoRA.** In principle, vanilla LoRA learns weight offsets relative to the base model, whereas the proposed input/output adaptation learns linear transformations in the feature space. These two mechanisms are complementary and can potentially be combined to harness both functionalities. The combined input and output adaptation can be implemented as:

$$Y = X((I + A_1 B_1)W + A_2 B_2) \quad \text{and} \quad Y = X(W(I + A_1 B_1) + A_2 B_2), \tag{6}$$

respectively. We conduct experiments on FLUX-1.dev and FLUX-1.schnell Labs (2024) using the output adaptation strategy to illustrate this effect. The total rank is fixed at 4, while we vary the proportion of rank allocated to each component. As shown in Fig. 7, the best `CLIP-T` performance is achieved when 25% of the rank is assigned to output adaptation. This extension allows flexible control over the trade-off between identity preservation and prompt alignment.

**Choices of Adaptation Layers.** In practice, prior knowledge of the target model can inform the choice of feature layers to adapt. For example, in the case of SD-XL Podell et al. (2023), if the target model—such as SDXL-Turbo Sauer et al. (2024)—is known to share the same text encoder for processing input prompts, it is advantageous to focus adaptation on the cross-attention layers, which primarily govern text-image interactions. The results in Tab. 1 (right) highlight this advantage: when all attention layers in SD-XL are fine-tuned, output adaptation slightly outperforms input adaptation. In contrast, when only cross-attention layers are made learnable, input adaptation achieves better performance, benefiting from the exact alignment of input feature spaces and yielding superior cross-model generalization.

## 6 CONCLUSIONS

In this paper, we identify and investigate a fundamental limitation of existing fine-tuning methods, such as LoRA: their unsatisfactory cross-model transferability due to the strong coupling between adaptation components and the base model weights. To overcome this issue, we propose simple yet effective alternatives—input and output adaptation—which decouple the adaptation from model-specific parameters and instead operate in the input/output feature spaces, by leveraging the underlying coherence of these spaces across parameter-wise model variants. Our approaches allow the adaptation components to be dynamically modulated by the target model's native weights, thereby significantly enhancing robustness in cross-model transfer scenarios. Our theoretical analysis supports the effectiveness of the proposed method and highlights its ability to model a broader class of linear transformations compared to traditional approaches. Notably, the proposed strategies require only one-line code changes and are compatible with a wide range of PEFT techniques. Extensive experiments across various models and tasks—such as personalized image generation, architectural adaptation, high-resolution adaptation, and large language models—demonstrate that our methods maintain competitive performance on the source model while consistently outperforming existing approaches in cross-model transfer settings.

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

In this part, we provide additional information related to the proposed input/output feature adaptation methods that cannot fit into the main manuscript due to the page limit, including theoretical proofs, additional implementation details, additional experimental results, the use of large language models (LLMs), and impacts, limitations, and future work.

## A  THEORETICAL PROOF

### A.1  ASSUMPTION OF CONSISTENT RELATIVE REPRESENTATION SIMILARITY

We first provide a theoretical proof to Prop. 1 and Prop. 2 of the main manuscript. For better readability, we summarize the theoretical setups and conclusions here again.

Consider one linear layer, and the source and target models can be written as $Y = XW$ and $Y' = XW'$, respectively. $x_1$ and $x_2$ are two feature vectors in the input space of the source model, and $y_1$ and $y_2$ are the corresponding output vectors with the parameter $W$. $x_1'$, $x_2'$, $y_1'$, and $y_2'$ are defined for the target model, similarly. Assume that the cosine similarity between $x_1$ and $x_2$, denoted as $\mathrm{sim}(x_1, x_2)$, is equal to that between $x_1'$ and $x_2'$, *i.e.*, $\mathrm{sim}(x_1, x_2) = \mathrm{sim}(x_1', x_2')$, and that $s_0 := \mathrm{sim}(y_1, y_2) = \mathrm{sim}(y_1', y_2')$. The three adaptation strategies in Eqs. 1 and 3 are considered. The adapters $AB$ are trained on the source model and transferred to the target. Assume that the adaptation components are $L2$-regularized, *i.e.*, $\|AB\|_2^2 \le \varepsilon$. We are interested in the similarity distance between output vectors produced under various adaptation strategies, *i.e.*, $|\mathrm{sim}(\hat{y}_1, \hat{y}_2) - \mathrm{sim}(\hat{y}_1', \hat{y}_2')|$, where $\hat{\ }$ denotes outputs from models after adaptation. The theoretical results concerning the three strategies are as follows:

**Lemma 1** (First-order expansion of cosine similarity). *Let $\mathbf{u}, \mathbf{v} \in \mathbb{R}^d$ with $\|\mathbf{u}\| > 0, \|\mathbf{v}\| > 0$ and*

$$s(\mathbf{u}, \mathbf{v}) := \frac{\mathbf{u}^\top \mathbf{v}}{\|\mathbf{u}\|\|\mathbf{v}\|}.$$

*For adaptation components $\Delta \mathbf{u}, \Delta \mathbf{v} \in \mathbb{R}^d$ one has the exact first-order expansion*

$$s(\mathbf{u} + \Delta \mathbf{u}, \mathbf{v} + \Delta \mathbf{v}) = s(\mathbf{u}, \mathbf{v}) + \delta + R,$$

*where the first-order term $\delta$ is*

$$\delta = \frac{\Delta \mathbf{u}^\top \mathbf{v} + \mathbf{u}^\top \Delta \mathbf{v}}{\|\mathbf{u}\|\|\mathbf{v}\|} - \frac{\mathbf{u}^\top \mathbf{v}}{\|\mathbf{u}\|^3 \|\mathbf{v}\|} \mathbf{u}^\top \Delta \mathbf{u} - \frac{\mathbf{u}^\top \mathbf{v}}{\|\mathbf{u}\|\|\mathbf{v}\|^3} \mathbf{v}^\top \Delta \mathbf{v}, \tag{7}$$

*and the remainder $R$ satisfies $R = O(\|\Delta \mathbf{u}\|^2 + \|\Delta \mathbf{v}\|^2)$. Moreover, the following convenient first-order norm bound holds:*

$$|\delta| \le (1 + |s(\mathbf{u}, \mathbf{v})|)\left(\frac{\|\Delta \mathbf{u}\|}{\|\mathbf{u}\|} + \frac{\|\Delta \mathbf{v}\|}{\|\mathbf{v}\|}\right). \tag{8}$$

*Proof.* Define $f(\mathbf{u}, \mathbf{v}) = \mathbf{u}^\top \mathbf{v}$ and $g(\mathbf{u}, \mathbf{v}) = \|\mathbf{u}\|\|\mathbf{v}\|$. Then $s = f/g$. Compute differentials:

$$\mathrm{d}f = \mathrm{d}\mathbf{u}^\top \mathbf{v} + \mathbf{u}^\top \mathrm{d}\mathbf{v},$$

and

$$\mathrm{d}g = \frac{\mathbf{u}^\top \mathrm{d}\mathbf{u}}{\|\mathbf{u}\|} \|\mathbf{v}\| + \|\mathbf{u}\| \frac{\mathbf{v}^\top \mathrm{d}\mathbf{v}}{\|\mathbf{v}\|}.$$

Using the quotient rule for differentials,

$$\mathrm{d}s = \frac{\mathrm{d}f \cdot g - f \cdot \mathrm{d}g}{g^2}.$$

Substitute $f = \mathbf{u}^\top \mathbf{v}$ and $g = \|\mathbf{u}\|\|\mathbf{v}\|$ and replace $\mathrm{d}\mathbf{u}, \mathrm{d}\mathbf{v}$ by specific increments $\Delta \mathbf{u}, \Delta \mathbf{v}$ to obtain the formula Eq. 7.

For the bound Eq. 8 apply Cauchy–Schwarz:

$$|\Delta \mathbf{u}^\top \mathbf{v}| \le \|\Delta \mathbf{u}\| \|\mathbf{v}\|, \qquad |\mathbf{u}^\top \Delta \mathbf{v}| \le \|\mathbf{u}\| \|\Delta \mathbf{v}\|.$$

Also

$$\left| \frac{\mathbf{u}^\top \mathbf{v}}{\|\mathbf{u}\|^3 \|\mathbf{v}\|} \mathbf{u}^\top \Delta \mathbf{u} \right| \le |s(\mathbf{u}, \mathbf{v})| \frac{\|\Delta \mathbf{u}\|}{\|\mathbf{u}\|}, \qquad \left| \frac{\mathbf{u}^\top \mathbf{v}}{\|\mathbf{u}\| \|\mathbf{v}\|^3} \mathbf{v}^\top \Delta \mathbf{v} \right| \le |s(\mathbf{u}, \mathbf{v})| \frac{\|\Delta \mathbf{v}\|}{\|\mathbf{v}\|}.$$

Combine these inequalities to get Eq. 8.

The statement that $R = O(\|\Delta \mathbf{u}\|^2 + \|\Delta \mathbf{v}\|^2)$ follows from the fact that $s$ is $C^\infty$ on the open set $\{(\mathbf{u}, \mathbf{v}) : \|\mathbf{u}\| > 0, \|\mathbf{v}\| > 0\}$, hence Taylor expansion with remainder applies. $\qquad \square$

To control the remainder quantitatively, we will use a Hessian (second derivative) bound. Write the gradient components

$$\nabla_{\mathbf{u}} s(\mathbf{u}, \mathbf{v}) = \frac{\mathbf{v}}{\|\mathbf{u}\| \|\mathbf{v}\|} - s(\mathbf{u}, \mathbf{v}) \frac{\mathbf{u}}{\|\mathbf{u}\|^2}, \qquad \nabla_{\mathbf{v}} s(\mathbf{u}, \mathbf{v}) = \frac{\mathbf{u}}{\|\mathbf{u}\| \|\mathbf{v}\|} - s(\mathbf{u}, \mathbf{v}) \frac{\mathbf{v}}{\|\mathbf{v}\|^2}.$$

Denote by $H(\mathbf{u}, \mathbf{v})$ the block Hessian of $s$ (a $2d \times 2d$ matrix with blocks $H_{uu}, H_{uv}, H_{vu}, H_{vv}$). A direct (but routine) calculation of these second derivatives yields that each block can be bounded in operator norm in terms of $\|\mathbf{u}\|, \|\mathbf{v}\|$ and $|s(\mathbf{u}, \mathbf{v})|$. A simple coarse (but explicit) bound that suffices for our purpose is:

**Lemma 2** (Hessian operator-norm bound). *Let $m := \min(\|\mathbf{u}\|, \|\mathbf{v}\|) > 0$. Then for the Hessian $H(\mathbf{u}, \mathbf{v})$ of $s$ at $(\mathbf{u}, \mathbf{v})$ one has the operator-norm bound*

$$\|H(\mathbf{u}, \mathbf{v})\|_{\mathrm{op}} \le \frac{C(1 + |s(\mathbf{u}, \mathbf{v})|)}{m^2},$$

*where one may take e.g. $C = 12$ (a conservative explicit constant).*

*Sketch of proof and explanation of the constant.* We only sketch the elementary (but somewhat tedious) steps. Differentiate the explicit formulae for $\nabla_{\mathbf{u}} s$ and $\nabla_{\mathbf{v}} s$. Each component of the block derivatives is a linear combination of terms of the following types (schematically):

$$\frac{\cdot}{\|\mathbf{u}\|^2}, \quad \frac{\cdot}{\|\mathbf{v}\|^2}, \quad \frac{\cdot}{\|\mathbf{u}\| \|\mathbf{v}\|}, \quad \frac{\cdot}{\|\mathbf{u}\|^3 \|\mathbf{v}\|}, \quad \text{etc.}$$

Each "$\cdot$" is either a vector of norm $\le \|\mathbf{u}\|$ or $\|\mathbf{v}\|$ or the scalar $s(\mathbf{u}, \mathbf{v})$. By repeatedly applying Cauchy–Schwarz and triangle inequality, one obtains bounds for each block's operator norm by expressions of the form $C_i(1 + |s|)/m^2$. Gathering all blocks and using subadditivity of operator norms yields the stated bound. Choosing $C = 12$ is conservative and covers the numerical coefficients arising from all block contributions. $\qquad \square$

Using the integral form of the remainder in Taylor's theorem (or the mean-value form for vector-valued functions), we obtain a quantitative second-order remainder bound: for increments $\Delta \mathbf{u}, \Delta \mathbf{v}$,

$$|R| \le \frac{1}{2} \sup_{t \in [0,1]} \|H(\mathbf{u} + t\Delta \mathbf{u}, \mathbf{v} + t\Delta \mathbf{v})\|_{\mathrm{op}} \left( \|\Delta \mathbf{u}\|^2 + \|\Delta \mathbf{v}\|^2 \right).$$

Applying Lemma 2 to the point $(\mathbf{u} + t\Delta \mathbf{u}, \mathbf{v} + t\Delta \mathbf{v})$ gives an explicit $O(\|\Delta \mathbf{u}\|^2 + \|\Delta \mathbf{v}\|^2)$ estimate whose constant depends on the minimum norms along the segment.

**Proposition 1.** *After applying the vanilla LoRA shown in Eq. 1 on both source and target models, the similarity distance between output vectors satisfies $|\mathrm{sim}(\hat{y}_1, \hat{y}_2) - \mathrm{sim}(\hat{y}'_1, \hat{y}'_2)| \le (1 + s_0)\Big( \frac{\|\mathbf{x}_1\| \varepsilon}{\|\mathbf{x}_1 W\|} + \frac{\|\mathbf{x}_2\| \varepsilon}{\|\mathbf{x}_2 W\|} + \frac{\|\mathbf{x}'_1\| \varepsilon}{\|\mathbf{x}'_1 W'\|} + \frac{\|\mathbf{x}'_2\| \varepsilon}{\|\mathbf{x}'_2 W'\|} \Big) + O(\varepsilon^2).$*

*Proof.* Apply Lemma 1 to the pair $(y_1, y_2)$ with

$$\Delta y_1 = x_1 AB, \qquad \Delta y_2 = x_2 AB.$$

We obtain the expansion

$$s(\hat{y}_1, \hat{y}_2) = s(y_1, y_2) + \delta_1 + R_1,$$

with $\delta_1$ given by Eq. 7 (with $(\mathbf{u}, \mathbf{v}) = (y_1, y_2)$ and $\Delta \mathbf{u} = x_1 AB, \Delta \mathbf{v} = x_2 AB$) and $|R_1| \le \frac{1}{2} \sup_{t \in [0,1]} \|H(y_1 + t\Delta y_1, y_2 + t\Delta y_2)\|_{\mathrm{op}} (\|\Delta y_1\|^2 + \|\Delta y_2\|^2)$.

Using Eq. 8 and $\|\Delta y_i\| = \|x_i AB\| \leq \|x_i\| \|AB\|_2 \leq \|x_i\| \varepsilon$ we get

$$|\delta_1| \leq (1 + s_0) \Big( \frac{\|x_1\| \varepsilon}{\|y_1\|} + \frac{\|x_2\| \varepsilon}{\|y_2\|} \Big).$$

For the remainder $R_1$ apply Lemma 2 at the segment points: since every point on the segment $\{(y_1 + t\Delta y_1, y_2 + t\Delta y_2) : t \in [0,1]\}$ has norms bounded below by $A_{\min} > 0$, we have

$$|R_1| \leq \frac{1}{2} \cdot \frac{C'(1 + |s|_{\max})}{A_{\min}^2} \big( \|\Delta y_1\|^2 + \|\Delta y_2\|^2 \big) \leq \frac{C'}{2} A_{\min}^{-2} (\|x_1\|^2 + \|x_2\|^2) \, \varepsilon^2,$$

where $|s|_{\max}$ denotes the maximum of $|s(\cdot, \cdot)|$ along the segment; we can bound $1 + |s|_{\max} \leq 2$ conservatively but here absorb it into $C'$.

An identical argument applies for the primed pair, giving $|\delta_2| \leq (1 + s_0)(\|x_1'\| \varepsilon / \|y_1'\| + \|x_2'\| \varepsilon / \|y_2'\|)$ and a remainder $R_2$ bounded by the same style of $\varepsilon^2$ term (with $\|x_1'\|^2 + \|x_2'\|^2$). By the triangle inequality,

$$\big| s(\hat{y}_1, \hat{y}_2) - s(\hat{y}_1', \hat{y}_2') \big| \leq |\delta_1| + |\delta_2| + |R_1| + |R_2|,$$

which yields Prop. 1 after combining the $\varepsilon^2$-terms. □

**Remark A.1.** *The displayed bound separates the linear-in-$\varepsilon$ contribution (explicit and often dominant when $\varepsilon \ll 1$) and an explicit quadratic term in $\varepsilon^2$. Both parts are computable from the data.*

**Proposition 2.** *After applying the input or output adaptation shown in Eq. 3 on both source and target models, the similarity distance between output vectors satisfies $|\sin(\hat{y}_1, \hat{y}_2) - \sin(\hat{y}_1', \hat{y}_2')| \leq 4(1 + s_0)\varepsilon + O(\varepsilon^2)$.*

*Proof.* For input adaptation, it follows the same route as Prop. 1. Now the adaptation components are

$$\Delta y_1 = x_1 ABW, \quad \Delta y_2 = x_2 ABW,$$

with $\|\Delta y_1\| \leq \|x_1\| \|AB\|_2 \|W\| \leq \|x_1\| \varepsilon \|W\|$ (since $\|AB\|_2 \leq \|AB\|_F = \varepsilon$). Insert these bounds into Lemma 1 to get the linear-in-$\varepsilon$ terms; the second-order remainder is handled via Lemma 2 as before, producing an explicit $O(\varepsilon^2)$ term depending on $\|x_i\|, \|W\|$ and the minimal norms of $y_i, y_i'$.

For output adaptation, $\Delta y_i = y_i AB$ and $\|\Delta y_i\| \leq \|y_i\| \varepsilon$. Apply Lemma 1 to each pair to obtain a linear term bounded by $(1 + s_0)(\|y_1\| \varepsilon / \|y_1\| + \|y_2\| \varepsilon / \|y_2\|)$ for the first pair and similarly for the primed pair. Summing and controlling remainders as before yields the stated inequality. □

## A.2 ASSUMPTION OF CONSISTENT INPUT/OUTPUT FEATURE SPACES

We consider compositional linear models, *i.e.*, $Y = XW_1 W_2$, as base models and low-rank adaptation for our theoretical analysis. Assume that the data, represented as row vectors, are drawn from a $c$-dimensional Gaussian distribution with zero mean and covariance matrix $\Sigma = \sigma^2 I$, *i.e.*, $x \sim \mathcal{N}(0, \sigma^2 I)$. The adapters are trained on a source model, parameterized by $(W_1, W_2)$, and tested on a target model, parameterized by $(W_1', W_2')$. The overall functionalities of the source and target models are assumed to be similar, with their difference upper bounded in Frobenius norm, *i.e.*, $\|W_1 W_2 - W_1' W_2'\|_F^2 \leq \varepsilon$.

We then consider 3 adaptation strategies: vanilla LoRA, input adaptation, and output adaptation, applied to either $W_1$ or $W_2$[1], $W_1$, and $W_2$, respectively. The corresponding models are:

$$Y = X(W_1 + AB)W_2, \quad Y = X(I + AB)W_1 W_2, \quad \text{and} \quad Y = XW_1 W_2(I + AB). \quad (9)$$

We are interested in the output discrepancy between the source and target models after the adapters are trained on the source model, and summarize the theoretical results as follows:

**Lemma 3.** *The expectation of the output distance between the source and target models before fine-tuning satisfies $\mathbb{E}_{x \sim \mathcal{N}(0, \sigma^2 I)}[\|xW_1 W_2 - xW_1' W_2'\|_2^2] \leq \sigma^2 \varepsilon$.*

---

[1]Without loss of generality, we apply LoRA to $W_1$, as the analysis for the case of $W_2$ follows analogously.

*Proof.* Assume that the random vector $x \sim \mathcal{N}(0, \sigma^2 I)$, and consider two composite linear transformations: $W_1 W_2$ and $W_1' W_2'$, where the matrices have compatible dimensions. Define:

$$\Delta = W_1 W_2 - W_1' W_2'.$$

Then,

$$x W_1 W_2 - x W_1' W_2' = x \Delta.$$

We aim to compute the expected squared Euclidean norm:

$$\mathbb{E}\left[\|x\Delta\|_2^2\right].$$

Since $x \sim \mathcal{N}(0, \sigma^2 I)$, and using the known formula for the second moment of a quadratic form of a Gaussian:

$$\mathbb{E}\left[x A A^\top x^\top\right] = \sigma^2 \operatorname{tr}(A A^\top) = \sigma^2 \|A\|_F^2,$$

we get:

$$\mathbb{E}\left[\|x\Delta\|_2^2\right] = \sigma^2 \|\Delta\|_F^2.$$

Now, assume that:

$$\|W_1 W_2 - W_1' W_2'\|_F^2 = \|\Delta\|_F^2 = \varepsilon.$$

Then the final result is:

$$\boxed{\mathbb{E}\left[\|x W_1 W_2 - x W_1' W_2'\|_2^2\right] = \sigma^2 \varepsilon.}$$

$\square$

**Proposition 3.** *After applying the vanilla LoRA on both source and target models, the expectation of the output distance between them satisfies* $\mathbb{E}_{x \sim \mathcal{N}(0,\sigma^2 I)}[\|x(W_1 + AB)W_2 - x(W_1' + AB)W_2'\|_2^2] \leq \sigma^2(\sqrt{\varepsilon} + \|AB\|_2 \|W_2 - W_2'\|_F)^2$.

*Proof.* Assume a random vector $x \sim \mathcal{N}(0, \sigma^2 I)$, and let $AB$ be a known fixed linear transformation, where $A$ and $B$ are matrices with compatible dimensions.

We aim to compute the expected squared distance between two transformations:

$$x(W_1 + AB)W_2 \quad \text{and} \quad x(W_1' + AB)W_2'.$$

Define:

$$\Delta = (W_1 + AB)W_2 - (W_1' + AB)W_2' = (W_1 W_2 - W_1' W_2') + AB(W_2 - W_2').$$

Let $x \sim \mathcal{N}(0, \sigma^2 I)$, and let $\Delta$ be any deterministic matrix. Then:

$$\mathbb{E}\left[\|x\Delta\|_2^2\right] = \sigma^2 \|\Delta\|_F^2.$$

Using Lemma 3:

$$\mathbb{E}\left[\|x(W_1 + AB)W_2 - x(W_1' + AB)W_2'\|_2^2\right] = \sigma^2 \left\|(W_1 W_2 - W_1' W_2') + AB(W_2 - W_2')\right\|_F^2.$$

If $\|W_1 W_2 - W_1' W_2'\|_F^2 = \varepsilon$, we can upper-bound:

$$\|(W_1 W_2 - W_1' W_2') + AB(W_2 - W_2')\|_F \leq \|W_1 W_2 - W_1' W_2'\|_F + \|AB\|_2 \cdot \|W_2 - W_2'\|_F,$$

which gives:

$$\boxed{\mathbb{E}\left[\|x(W_1 + AB)W_2 - x(W_1' + AB)W_2'\|_2^2\right] \leq \sigma^2 \left(\sqrt{\varepsilon} + \|AB\|_2 \|W_2 - W_2'\|_F\right)^2.}$$

$\square$

**Proposition 4.** *After applying the input adaptation* $Y = X(I + AB)W_1 W_2$ *on both source and target models, the expectation of the output distance between them satisfies* $\mathbb{E}_{x \sim \mathcal{N}(0,\sigma^2 I)}[\|x(I + AB)W_1 W_2 - x(I + AB)W_1' W_2'\|_2^2] \leq \sigma^2 \|I + AB\|_2^2 \varepsilon$.

*Proof.* Assume a random vector $x \sim \mathcal{N}(0, \sigma^2 I)$, and a fixed known linear transformation $T = I + AB$, where $A$ and $B$ are matrices of compatible dimensions.

Consider two composite linear transformations: $W_1 W_2$ and $W_1' W_2'$, with

$$\Delta = W_1 W_2 - W_1' W_2'.$$

Then,

$$xTW_1 W_2 - xTW_1' W_2' = xT\Delta.$$

Let $x \sim \mathcal{N}(0, \sigma^2 I)$, and let $\Delta$ be a deterministic matrix. Then:

$$\mathbb{E}\left[\|x\Delta\|_2^2\right] = \sigma^2 \|\Delta\|_F^2.$$

Using Lemma 3 with $\Delta_T = T\Delta = (I + AB)(W_1 W_2 - W_1' W_2')$, we have:

$$\mathbb{E}\left[\|x(I + AB)W_1 W_2 - x(I + AB)W_1' W_2'\|_2^2\right] = \mathbb{E}\left[\|x\Delta_T\|_2^2\right] = \sigma^2 \|(I+AB)(W_1 W_2 - W_1' W_2')\|_F^2.$$

Using the submultiplicativity of the Frobenius norm:

$$\|(I + AB)(W_1 W_2 - W_1' W_2')\|_F \leq \|I + AB\|_2 \cdot \|W_1 W_2 - W_1' W_2'\|_F = \|I + AB\|_2 \cdot \sqrt{\varepsilon}.$$

Thus, we obtain the upper bound:

$$\boxed{\mathbb{E}\left[\|x(I + AB)W_1 W_2 - x(I + AB)W_1' W_2'\|_2^2\right] \leq \sigma^2 \|I + AB\|_2^2 \cdot \varepsilon.}$$

$\square$

**Proposition 5.** *After applying the output adaptation $Y = XW_1 W_2(I + AB)$ on both source and target models, the expectation of the output distance between them satisfies $\mathbb{E}_{x \sim \mathcal{N}(0, \sigma^2 I)}[\|xW_1 W_2(I + AB) - xW_1' W_2'(I + AB)\|_2^2] \leq \sigma^2 \|I + AB\|_2^2 \varepsilon$.*

*Proof.* Assume a random vector $x \sim \mathcal{N}(0, \sigma^2 I)$, and let $T = I + AB$ be a known fixed linear transformation, where $A$ and $B$ are matrices of compatible dimensions.

Consider two composite transformations:

$$xW_1 W_2 T \quad \text{and} \quad xW_1' W_2' T.$$

Define the difference:

$$\Delta = W_1 W_2 - W_1' W_2',$$

so that:

$$xW_1 W_2 T - xW_1' W_2' T = x\Delta T.$$

Let $x \sim \mathcal{N}(0, \sigma^2 I)$, and let $\Delta \in \mathbb{R}^{d \times m}$, then:

$$\mathbb{E}\left[\|x\Delta\|_2^2\right] = \sigma^2 \|\Delta\|_F^2.$$

Apply Lemma 3 to $\Delta_T = \Delta T = (W_1 W_2 - W_1' W_2')(I + AB)$, yielding:

$$\mathbb{E}\left[\|xW_1 W_2(I + AB) - xW_1' W_2'(I + AB)\|_2^2\right] = \mathbb{E}\left[\|x\Delta_T\|_2^2\right] = \sigma^2 \|(W_1 W_2 - W_1' W_2')(I+AB)\|_F^2.$$

Using the submultiplicativity of the Frobenius norm:

$$\|\Delta T\|_F \leq \|\Delta\|_F \cdot \|T\|_2 = \sqrt{\varepsilon} \cdot \|I + AB\|_2,$$

we obtain the bound:

$$\boxed{\mathbb{E}\left[\|xW_1 W_2(I + AB) - xW_1' W_2'(I + AB)\|_2^2\right] \leq \sigma^2 \|I + AB\|_2^2 \cdot \varepsilon.}$$

$\square$

Intuitively, the above propositions suggest that when two models exhibit similar overall functionality, their outputs—after applying the proposed input/output adapters—also tend to be similar, regardless of discrepancies at the individual layer level. These properties enhance their cross-model transferability across various parameter-wise model variants. We include a toy example to further illustrate these effects in Fig. 8.

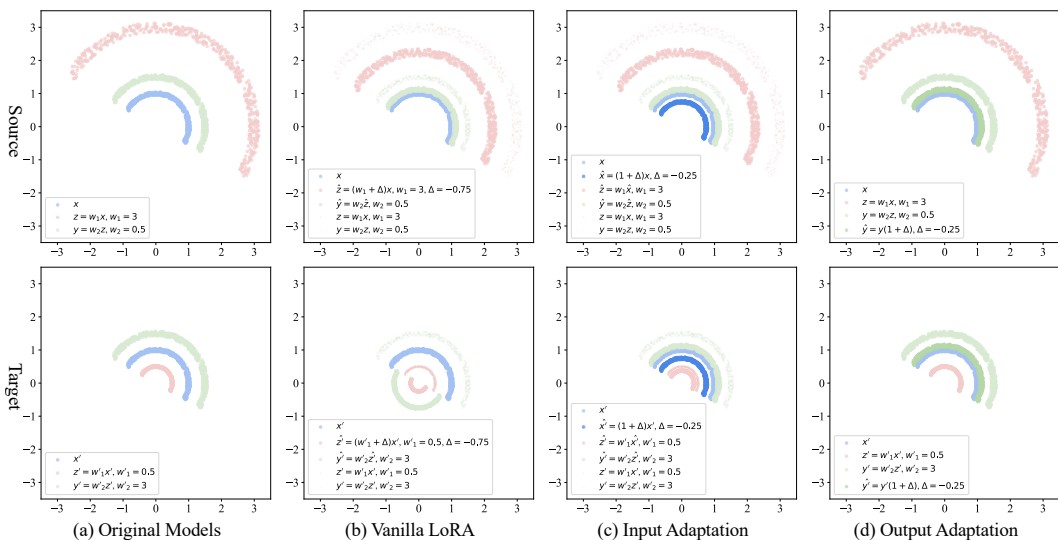

(a) Original Models    (b) Vanilla LoRA    (c) Input Adaptation    (d) Output Adaptation

Figure 8: A toy example illustrating the theoretical analysis of vanilla LoRA and the proposed input/output adaptations. The sample follows a 2D "crescent" distribution. The source and target models are composed of affine transformations with the same overall functionality but different parameters. Blue, pink, and green points represent the inputs, intermediate features, and outputs, respectively. In this setting, vanilla LoRA fails to preserve output consistency after adaptation, whereas the proposed input/output adaptation demonstrates greater robustness.

| Model | SD-v1.5 | SD-v3.5-Large | FLUX-1.dev | SD-XL | SD-XL (Cross-Att.) |
|---|---|---|---|---|---|
| Optimizer | AdamW | Prodigy | Prodigy | AdamW | AdamW |
| Learning Rate | 1e-4 | 1 | 1 | 1e-4 | 1e-4 |
| Batch Size | | | 4 | | |
| Training Iterations | | | 500 | | |
| Rank | | LoRA: 4 / DoRA: 4 / LoHa: 64 / LoKr: 128 | | | |
| Resolution | | | 512 | | |
| Target Modules | ["to_k",
"to_q",
"to_v",
"to_out.0",
"add_k_proj",
"add_v_proj"] | ["to_k",
"to_q",
"to_v",
"to_out.0",
"add_k_proj",
"add_v_proj",
"add_q_proj",
"to_add_out"] | ["to_k",
"to_q",
"to_v",
"to_out.0",
"add_k_proj",
"add_v_proj",
"add_q_proj",
"to_add_out"] | ["to_k",
"to_q",
"to_v",
"to_out.0"] | ["attn2.to_k",
"attn2.to_v"] |
| Hardware | | | 1×RTX6000 Ada | | |

Table 7: Detailed configurations of hyperparameters and training setups for the experiments of fine-tuning-based personalized image generation.

# B ADDITIONAL IMPLEMENTATION DETAILS

For reproducibility, we provide detailed configurations of experiments in the main manuscript.

**Input/Output Adaptation.** The proposed input/output adaptation strategies are implemented by modifying the source codes of the popular PEFT library[2]. The code is under the Apache License Version 2.0. Specifically, the codes for defining the shapes of the adapters, forward propagation, and merging the adapter weights with the original ones are modified. The modified files are attached as supplementary materials.

---

[2]https://github.com/huggingface/peft

**Evaluation.** The main evaluation is conducted on the DreamBooth Ruiz et al. (2023) dataset[3]. The dataset is under the license of Creative Commons Attribution 4.0 International, with complete licenses for all the included images.

**Fine-Tuning-Based Personalized Image Generation.** The experiments of fine-tuning-based personalized image generation are conducted on the popular codebase provided by the DreamBooth example in the Diffusers library[4]. The code is under the Apache License Version 2.0. We experiment with four models: SD-v1.5, SD-v3.5-Large, FLUX-1.dev, and SD-XL. Following convention, we mainly add adapters to the weights in the attention layers. The detailed configurations like optimizer, learning rate, batch size, *etc.* are provided in Tab. 7.

**Adapter-Based Personalized Image Generation.** The idea of adapter-based personalized image generation is to tame an adapter that maps a subject image into the conditional space of a diffusion model. For example, for SD-v1.5, the cross-attention layers are adopted to handle the interactions between image features and text conditions. A series of methods like ELITE Wei et al. (2023) train an image encoder that encodes a subject image into multiple key-value tokens used for cross-attention. It is demonstrated that optimizing additional key-value mapping parameters for tokens from subject images can benefit performance. However, since the dimensions and configurations of cross-attention layers in various diffusion models are different, the optimized key-value mapping parameters in one model are inapplicable to other unseen models in general. To handle this drawback, in this paper, we deploy the proposed input adaptation strategy to this field, since the text embedding spaces in a series of diffusion models are aligned: multiple diffusion models like the Stable Diffusion series adopt the CLIP text encoder to encode input prompts.

Specifically, we build the input adaptation strategy based on ELITE Wei et al. (2023) due to its availability of codes for both training and inference[5]. The code is under the Apache License Version 2.0. Instead of introducing additional key and value mapping parameters for subject features, we use the native parameters in the cross-attention layers to handle them. Moreover, we train a mapping network to encode the subject features into the word embedding space of the CLIP text encoder and fine-tune the text encoder as well. In this way, the input for diffusion models, *i.e.*, CLIP text features, are adapted to contain subject-relevant features.

We train the mapping network and the text encoder for 80,000 iterations on 4 RTX6000 Ada GPUs, which takes around 1 day. Other configurations, including training data, optimization, the structure of the encoder, *etc.*, follow the default setups of ELITE. SD-v1.5 is adopted in training, while distilled SD, SD-XL, and SD-3 are adopted in evaluation.

**Controllable Image Generation.** Recent works like OminiControl Tan et al. (2024) achieve controllable image generation through LoRA. By taming a low-rank adapter for each control signal and concatenating tokens from the control signal with the latent tokens of a pre-trained diffusion transformer, the generated results are trained to follow the input conditions. In this paper, we build the proposed input/output adaptation strategies on OminiControl[6] with the Canny edge condition. The code is under the Apache License Version 2.0. The training is conducted on 2 H100 GPUs for 15,000 steps. Other configurations maintain the same as the original OminiControl. FLUX-1.dev and FLUX-1.schnell are adopted as training and evaluation models, respectively.

**Architectural Adaptation.** Recent works like CLEAR Liu et al. (2024c) demonstrate that it is feasible to replace self-attention in a pre-trained diffusion transformer, which results in high inference latency, especially for high-resolution images, with efficient alternatives like neighborhood attention. After a distillation-based adaptation process, it turns out that the efficient counterpart achieves performance comparable to or even better than the original model.

In this work, we adopt the "undistilled" FLUX-1.dev model[7] as the teacher model seen during training and evaluate the adaptation components on FLUX-1.dev. Following the default setup of CLEAR[8], which is under the Apache License Version 2.0, the entire `to_q`, `to_k`, `to_v`, and `to_out`

---

[3]https://github.com/google/dreambooth/tree/main/dataset

[4]https://github.com/huggingface/diffusers/tree/main/examples/dreambooth

[5]https://github.com/csyxwei/ELITE

[6]https://github.com/Yuanshi9815/OminiControl

[7]https://huggingface.co/ashen0209/Flux-Dev2Pro

[8]https://github.com/Huage001/CLEAR

| PEFT | Model | SD-v1.5 | | | SD-XL | | | SD-v3.5 | | | FLUX | | |
|---|---|---|---|---|---|---|---|---|---|---|---|---|---|
| | Metric | C-T | C-I | D-I | C-T | C-I | D-I | C-T | C-I | D-I | C-T | C-I | D-I |
| LoRA | Source Model | **.294** | .770 | .578 | .293 | **.815** | **.694** | .307 | .787 | .652 | .298 | .801 | .679 |
| | | .293 | .777 | .587 | .299 | .806 | .671 | **.308** | .785 | .645 | **.299** | .797 | .666 |
| | | **.294** | **.790** | **.626** | **.300** | .811 | .690 | .306 | **.794** | **.665** | .296 | **.808** | **.685** |
| | Target Model | .289 | .785 | .608 | .309 | .752 | .592 | .301 | .780 | .642 | **.308** | .773 | .626 |
| | | .288 | .792 | .618 | **.316** | .759 | .592 | **.302** | .782 | .642 | **.308** | .784 | .627 |
| | | **.294** | **.793** | **.634** | .309 | **.762** | **.614** | .298 | **.794** | **.659** | .306 | **.789** | **.639** |
| DoRA | Source Model | **.294** | .772 | .580 | .293 | **.815** | **.693** | .308 | .782 | .644 | **.303** | .801 | .672 |
| | | .284 | .786 | .609 | .291 | **.815** | .690 | **.311** | .787 | .651 | **.303** | .803 | .676 |
| | | **.294** | **.791** | **.627** | **.300** | .810 | .689 | .305 | **.795** | **.665** | .300 | **.810** | **.689** |
| | Target Model | .290 | .784 | .607 | **.322** | .735 | .538 | .303 | .778 | .642 | **.314** | .772 | .612 |
| | | .289 | .790 | .615 | .317 | **.751** | **.569** | **.307** | .784 | .648 | .311 | .775 | .613 |
| | | **.294** | **.791** | **.633** | .321 | .743 | .557 | .299 | **.793** | **.656** | .308 | **.781** | **.633** |
| LoHa | Source Model | .301 | .759 | .550 | **.300** | .808 | **.677** | .301 | **.814** | **.693** | .275 | .819 | .697 |
| | | .302 | .760 | .553 | .297 | .810 | .676 | **.302** | .811 | .688 | .279 | **.820** | **.700** |
| | | **.303** | **.774** | **.595** | **.300** | **.811** | .671 | **.302** | .809 | .683 | **.281** | .816 | .685 |
| | Target Model | .304 | .698 | .429 | **.317** | .756 | .591 | .289 | **.810** | **.690** | .288 | .797 | .658 |
| | | .304 | **.709** | **.443** | .312 | **.767** | **.607** | .290 | **.810** | **.690** | **.294** | **.810** | **.683** |
| | | **.308** | .698 | .430 | .315 | **.767** | **.607** | .290 | .806 | .676 | **.294** | .804 | .664 |
| LoKr | Source Model | **.295** | .788 | .617 | .315 | .757 | .545 | .304 | .805 | .680 | .289 | .818 | .704 |
| | | **.295** | .790 | .621 | .312 | **.766** | **.560** | **.305** | .806 | .681 | **.291** | .810 | .682 |
| | | .291 | **.802** | **.648** | **.319** | .744 | .509 | .302 | **.813** | **.691** | .288 | **.820** | **.705** |
| | Target Model | **.300** | .759 | .560 | **.323** | .716 | .483 | **.294** | .801 | .676 | **.305** | .785 | .625 |
| | | .299 | .760 | .566 | .321 | **.726** | **.501** | **.294** | .802 | .678 | .304 | .786 | .630 |
| | | **.300** | **.770** | **.588** | **.323** | .718 | .486 | .291 | **.810** | **.687** | .302 | **.797** | **.659** |

Table 8: Full performance in same-model and cross-model settings for various PEFT methods and the proposed input/output adaptation strategies built upon each of them on the DreamBooth benchmark.

| LoRA Rank & Metric | 1 | | | 4 | | | 16 | | | 64 | | | 256 | | |
|---|---|---|---|---|---|---|---|---|---|---|---|---|---|---|---|
| | C-T | C-I | D-I | C-T | C-I | D-I | C-T | C-I | D-I | C-T | C-I | D-I | C-T | C-I | D-I |
| Param. Ada. | **.316** | .754 | .570 | **.308** | .773 | .626 | **.309** | .790 | .650 | **.311** | .777 | .621 | **.309** | .787 | .647 |
| Input Ada. | .313 | .756 | .570 | **.308** | .784 | .627 | .308 | **.794** | .653 | **.311** | .780 | .629 | .306 | .789 | .647 |
| Output Ada. | .313 | **.773** | **.620** | .306 | **.789** | **.639** | .306 | .791 | **.654** | .308 | **.793** | **.651** | .303 | **.797** | **.659** |

Table 9: Performance in cross-model settings for vanilla LoRA and the proposed input/output adaptation strategies on the DreamBooth benchmark. The adapters are trained on FLUX-1.dev and evaluated on FLUX-1.schnell.

matrices are learnable. Therefore, the input and output adaptation strategies are implemented as $Y = X(I + A)W$ and $Y = XW(I + A)$, respectively, where $A$ is learnable and initialized as zero matrices. The radius of the circular attention window is set as 8, and the down-sampling factor is 8. The training is conducted on 4 H100 GPUs for 40,000 steps. Other configurations maintain the same as the original CLEAR.

**Large Language Model.** In this work, we build the proposed input/output adaptation strategies on the codebases of DoRA Liu et al. (2024b)[9] under the Apache License Version 2.0 and PiSSA Meng et al. (2024)[10], for the experiments on the commonsense reasoning task and the mathematical problem-solving and coding tasks, respectively. All the training and evaluation protocols follow their default setups.

## C  ADDITIONAL EXPERIMENTAL RESULTS

**Full Results of Fine-Tuning-Based Personalized Image Generation on Various Models and PEFT Methods.** See Tab. 8. Please refer to Fig. 9 for more visualized results.

---

[9]https://github.com/NVlabs/DoRA
[10]https://github.com/GraphPKU/PiSSA

| Model & Metric | FLUX-1.dev (Source) | | | FLUX-1.schnell (Target) | | |
|---|---|---|---|---|---|---|
| | C-T | C-I | D-I | C-T | C-I | D-I |
| Vanilla LoRA | .298 | .801 | .679 | **.308** | .773 | .626 |
| X-LoRA | .295 | .793 | .662 | .307 | .777 | .616 |
| Input Ada. | **.299** | .797 | .666 | **.308** | .784 | .627 |
| Output Ada. | .296 | **.808** | **.685** | .306 | **.789** | **.639** |

Table 10: Comparisons with the most recent work X-LoRA Shi (2024). The adapters are trained on FLUX-1.dev and evaluated on FLUX-1.schnell.

| Model & Metric | FLUX-1.dev (Source) | | | FLUX-1.schnell (Target) | | |
|---|---|---|---|---|---|---|
| | C-T | C-I | D-I | C-T | C-I | D-I |
| Vanilla LoRA | .298±.001 | .801±.004 | .679±.006 | .308±.001 | .773±.006 | .626±.011 |
| Input Ada. | .299±.001 | .797±.005 | .666±.011 | .308±.002 | .784±.004 | .627±.005 |
| Diff | .001±.001 | -.004±.004 | -.013±.010 | .000±.001 | .011±.003 | .001±.006 |
| Output Ada. | .296±.001 | .808±.005 | .685±.008 | .306±.002 | .789±.005 | .639±.010 |
| Diff | -.002±.001 | .007±.002 | .006±.006 | -.002±.001 | .016±.002 | .013±.004 |

Table 11: Results of standard deviation of various random seeds. The adapters are trained on FLUX-1.dev and evaluated on FLUX-1.schnell.

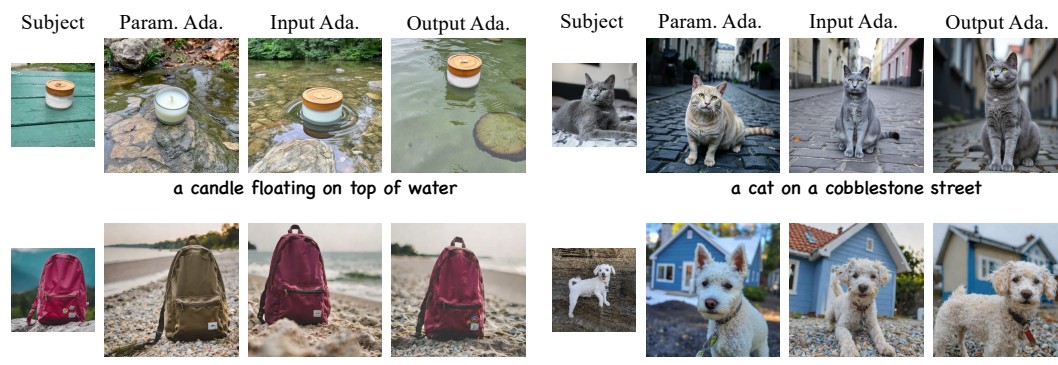

Subject  Param. Ada.  Input Ada.  Output Ada.   Subject  Param. Ada.  Input Ada.  Output Ada.

a candle floating on top of water          a cat on a cobblestone street

a backpack on the beach          a dog with a blue house in the background

Figure 9: More visualization of the proposed input/output adaptation strategies for fine-tuning-based personalized image generation.

**More Results of Adapter-Based Personalized Image Generation.** Seen on SD-v1.5, the large-to-small generalization results are shown in Fig. 10, and the small-to-large generalization results are shown in Fig. 11.

**Various LoRA Ranks.** See Tab. 9. The conclusion is insensitive to various LoRA ranks.

**Standard Deviations.** See Tab. 11. The proposed methods outperform the vanilla LoRA in terms of cross-model generalization on various random seeds consistently.

**Comparisons with the Most Recent Work.** We provide comparison results with X-LoRA Shi (2024), the most recent work on a similar topic. Due to the lack of the source code, we reproduce the algorithm given the implementation details from the original paper.

# D    THE USE OF LARGE LANGUAGE MODELS (LLMs)

In accordance with the ICLR 2026 policy on the use of large language models (LLMs) in paper writing, we disclose that LLMs were employed solely for language-related purposes. Specifically, we used LLMs to assist with grammar correction, sentence polishing, and improving the readability of the text. Importantly, LLMs were not used to generate novel research ideas or draw conclusions.

# E    IMPACTS, LIMITATIONS, AND FUTURE WORK

**Impacts.** If not properly regulated, the proposed input/output adaptation frameworks can potentially be used to generate information with negative social impacts, like fake news, illegal images, *etc.* Incorporating existing safe checkers, like the one used in the Stable Diffusion series, is helpful to alleviate these drawbacks.

**Limitations and Future Work.** Although this paper introduces the concepts of input and output adaptation, to keep the scope focused, we do not conduct fine-grained investigations to identify which layers or blocks in specific models benefit most from each type of adapter. Such explorations, tailored to specific cross-model transfer scenarios, are valuable directions for future work.

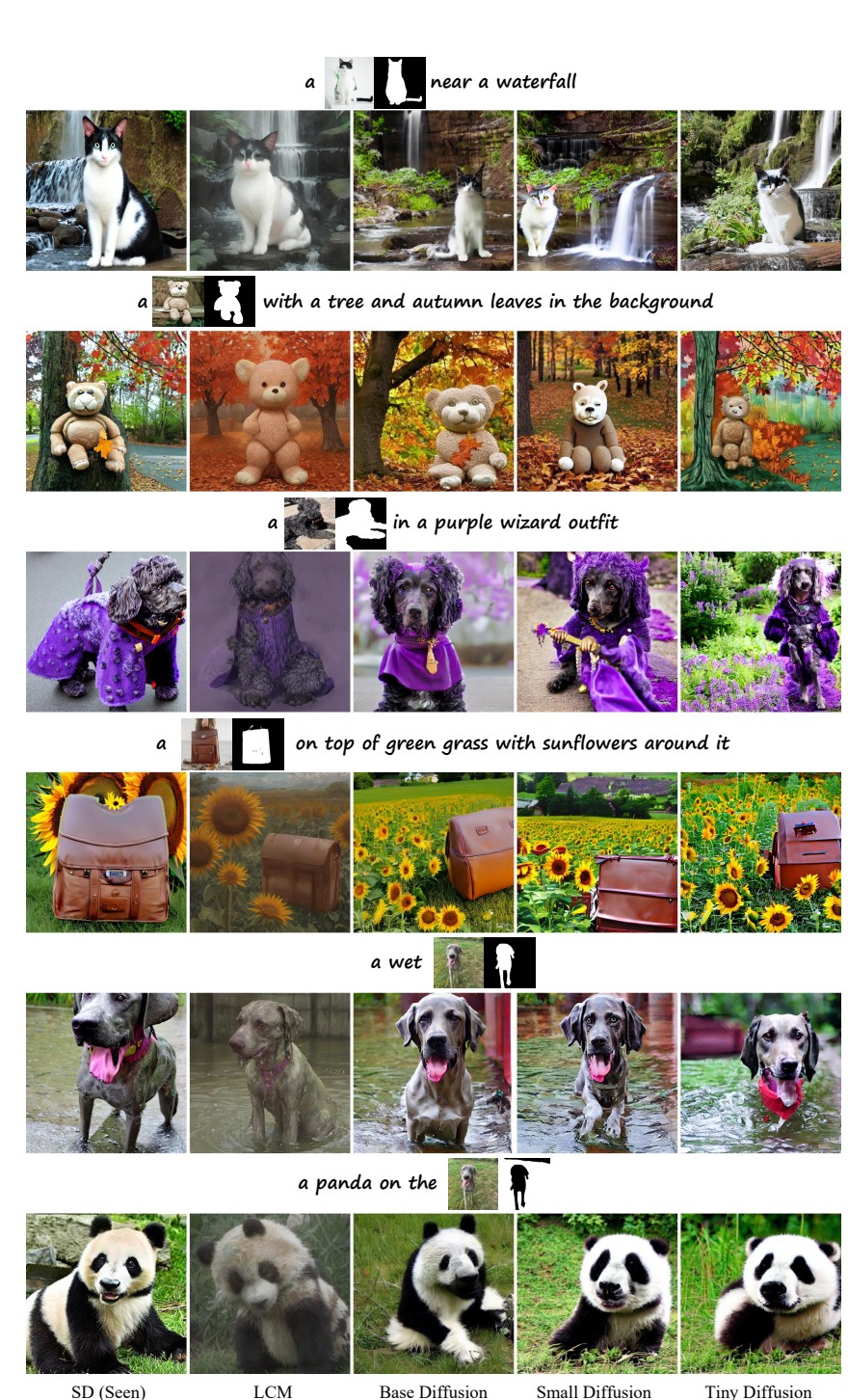

Figure 10: More visualization of the proposed input adaptation strategies for adapter-based personalized image generation. SD-v1.5 is the seen model during training, while the others are unseen.

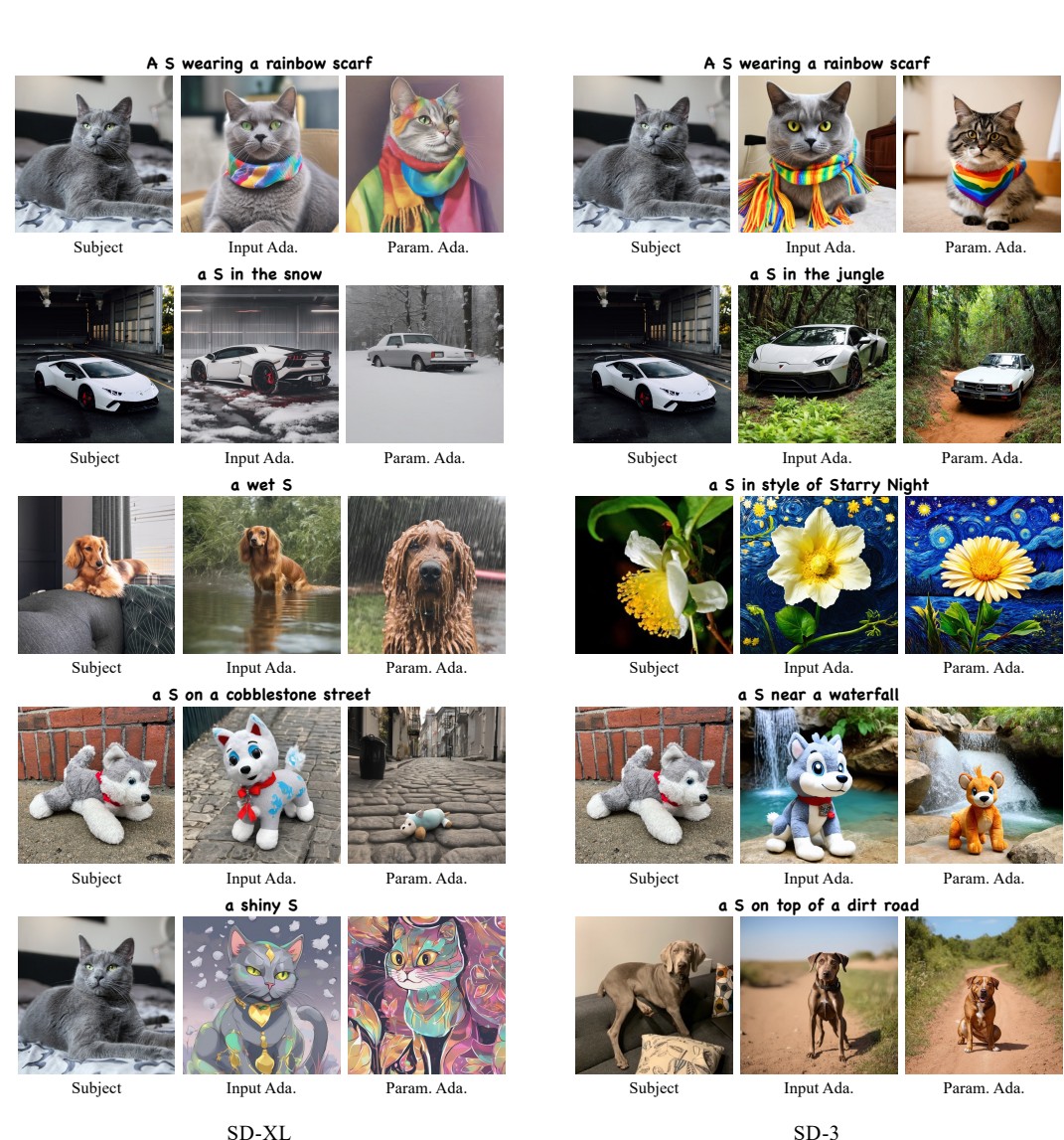

Figure 11: More visualization of the proposed input adaptation strategies for adapter-based personalized image generation. Results on two unseen models, SD-XL and SD-3, are shown here.

