# OpenReview forum: "Revisit Model Adaptation from Parameters to Features"
_ICLR.cc/2026/Conference — ICLR 2026 Conference Withdrawn Submission_

### Official Review · Reviewer_qEKi · 2025-10-15

**Soundness:** 3
**Presentation:** 3
**Contribution:** 3
**Rating:** 6
**Confidence:** 3

**Summary:**

This paper revisits the transferability of parameter-efficient fine-tuning (PEFT) methods, particularly LoRA, across models within the same family. The authors identify that existing fine-tuning adapters are tightly coupled to the base model’s parameters, making them fragile under weight-space transformations. To address this, the paper proposes two extremely simple yet effective strategies: input adaptation and output adaptation that operate directly in the feature space rather than in the parameter space.

**Strengths:**

1. Simplicity and generality.
The proposed input/output adaptation is strikingly simple: just reparameterize the adaptation to act before or after the base linear transformation. This design generalizes to other adapters (DoRA, LoHa, LoKr, etc.) and is compatible with existing PEFT frameworks with trivial integration.

2. Strong theoretical intuition.
The analysis (Propositions 1–2) formally shows that the similarity deviation upper bound for input/output adaptation is independent of the source and target model weights, explaining its robustness to model shifts.

3. Experiments span multiple domains including diffusion models, control adapters, architectural adaptation, and LLMs, each demonstrating consistent improvements. Tables 1–6 provide thorough quantitative evidence and align with qualitative visual results (Figs. 1, 4–7).

4. Ease of adoption and clarity.
The idea is clearly motivated, well-illustrated (Fig. 3), and practically appealing—requiring minimal code changes, no additional parameters, and no retraining overhead.

**Weaknesses:**

1. Limited discussion of failure cases.
The paper would benefit from an explicit analysis of when input/output adaptation fails and when feature spaces across models are not aligned, such as between architectures with disjoint tokenization or modality encoders.

2. Evaluation of scaling and cost.
All experiments are single-node or mid-size (≤ 32B for LLMs). It’s unclear how the approach scales to very large models (> 70B) or cross-architecture adaptation where feature-space consistency is weaker.

3. Theory–practice bridge could be tighter.
While the theoretical analysis is neat, its assumptions (e.g., consistent feature similarity) may not hold broadly. A synthetic or empirical validation of these assumptions would strengthen the connection between Section 4 and the experiments.

**Questions:**

See weaknesses

---

### Official Review · Reviewer_iofA · 2025-10-31

**Soundness:** 3
**Presentation:** 2
**Contribution:** 2
**Rating:** 4
**Confidence:** 2

**Summary:**

This paper investigates whether fine-tuning adapters like LoRA can be transferred across parameter-wise model variants. Theoretical analysis shows that strong coupling between adapters and base weights limits cross-model transfer and causes overfitting. To address this, the authors propose two decoupled adaptation methods operating on input and output features, enabling robust transfer by modulating adapters with the target model’s native parameters. These methods are plug-and-play and require minimal code changes. Experiments across diverse models show comparable source performance and superior transferability to existing approaches.

**Strengths:**

1. The paper is well-written and organized, with a clear structure.
2. The plug-and-play  method is novel.
3. The setup of the experiment is explained detailedly.

**Weaknesses:**

1. The theoretical analysis is quite simple, are there some discussions between the assumption you mentioned and the real world scenarios?
2. There should be more ablation studies. For instance, some choices (e.g., layer selection, interaction with normalization layers) could use further empirical clarification.
3. Can this method generalize to non-linear situations?

**Questions:**

See weakness.

---

### Official Review · Reviewer_6gRK · 2025-11-01

**Soundness:** 2
**Presentation:** 3
**Contribution:** 2
**Rating:** 4
**Confidence:** 4

**Summary:**

This paper analyzes why parameter-space adapters (e.g., LoRA) trained on one model often transfer poorly to variants, due to coupling between adapter parameters and base weights. The authors propose two simple alternatives: adaptation applied to input features and adaptation applied to output features, with an explicit decoupling scheme that aims to improve cross-model transferability. Theory is provided to explain transfer failure modes, and experiments (including DreamBooth-style evaluations) show that the proposed feature-space adaptations match source-model performance while improving cross-model transfer.

**Strengths:**

Solid theoretical diagnosis of coupling issues; derivations quantify how weight transformations can break adapters. Appendix contains formal statements and toy proofs (Gaussian assumptions used in derivations).

Practical, low-friction proposals (input/output feature adapters) that require minimal code changes and appear to keep source-model performance while improving transfer.

Extensive empirical evaluation across models and application settings; authors explicitly consider real-world transfer scenarios (e.g., cross-architecture, diffusion models).

**Weaknesses:**

The theoretical results rely on simplifying assumptions (Gaussian inputs, linearization) that limit generality; the paper would benefit from explicit discussion of these limitations and empirically verifying when assumptions are violated.

Some claimed experiments (cross-model transfer) need clearer protocol descriptions: are models fine-tuned with identical data and hyperparameters? How sensitive are results to optimizer/hyperparameter mismatch?

Incrementality: moving adaptation to features is not an significant enough research redesign - it’s an important engineering insight but may be considered incremental unless the paper more strongly highlights surprising theoretical or empirical phenomena.

**Questions:**

Provide a clearer mapping from the theoretical assumptions to practical settings: when (which architectures / input distributions) should the Gaussian/linearized approximations be considered valid?

How sensitive is cross-model transfer to minor implementation differences (e.g., layer normalization, tokenizer differences)? Please supply ablations.

Can the proposed feature adapters be combined with LoRA-like low-rank priors for further parameter savings, and if so, how does that affect transfer?

---

### Official Review · Reviewer_ghSp · 2025-11-01

**Soundness:** 2
**Presentation:** 1
**Contribution:** 2
**Rating:** 2
**Confidence:** 4

**Summary:**

This paper investigates why parameter-efficient fine-tuning methods such as LoRA struggle to transfer between models with different parameterizations. The authors show that these methods tightly couple adaptation components to model-specific weights, limiting cross-model generalization. To address this, authors propose simple input and output feature adaptation strategies that decouple adapters from base weights, improving transferability across models while maintaining competitive in-model performance.

**Strengths:**

1. The paper introduces an extension of low-rank adaptation by applying it in the input and output feature spaces rather than directly on model weights.
2. The proposed approach is conceptually simple yet theoretically well-motivated, offering a plug-and-play solution that enhances robustness across parameter variations.
3. The authors further validate their method through comprehensive experiments spanning both image generation and large language model tasks, demonstrating its broad applicability and consistent performance gains.

**Weaknesses:**

1. Novelty is questionable. The idea of adding additional adaptation layers (or modules) during fine-tuning has been explored extensively in prior work in parameter-efficient fine-tuning (PEFT) methods: for example, adapter modules and low-rank updates (e.g., Low‑Rank Adaptation (LoRA)) are well established. [1,2,3]
2. Marginal performance gains. In the results (e.g., Table 2 and Table 6) the improvement of the proposed method over vanilla LoRA appears limited; the paper does not appear to report statistical significance (e.g., error bars or variance across runs) which raises questions about robustness of the gains.
3. Theoretical clarity is weak. The argument for how the proposed method “decouples” from model weights compared to LoRA is vague — for example, equation 5 labelled “Conversion from Vanilla LoRA” suggests that LoRA might be seen theoretically convertible to the proposed method, thus weakening the claim of novelty or distinctness.
4. Poor presentation and formatting. The visual clarity and adherence to formatting guidelines are suboptimal. For example, Figure 7’s plots are too small to read clearly, and all table captions appear below rather than above the tables—contrary to ICLR 2026 formatting instructions. Additionally, required spacing before and after tables is missing, which detracts from overall readability and professionalism.

[1] AdapterFusion: Non-Destructive Task Composition for Transfer Learning

[2] AdaMix: Mixture-of-Adaptations for Parameter-efficient Model Tuning

[3] Parameter-efficient Multi-task Fine-tuning for Transformers via Shared Hypernetworks

**Questions:**

See Weaknesses.

---

### Note · Authors · 2025-11-14

I have read and agree with the venue's withdrawal policy on behalf of myself and my co-authors.